# Organotypic Culture of Testicular Tissue from Infant Boys with Cryptorchidism

**DOI:** 10.3390/ijms23147975

**Published:** 2022-07-19

**Authors:** Danyang Wang, Simone Hildorf, Elissavet Ntemou, Linn Salto Mamsen, Lihua Dong, Susanne Elisabeth Pors, Jens Fedder, Erik Clasen-Linde, Dina Cortes, Jørgen Thorup, Claus Yding Andersen

**Affiliations:** 1Laboratory of Reproductive Biology, University Hospital of Copenhagen, Rigshospitalet, 2100 Copenhagen, Denmark; elis.ntemou@gmail.com (E.N.); linn.salto.mamsen@regionh.dk (L.S.M.); lihua-dong@hotmail.com (L.D.); sup@gubra.dk (S.E.P.); claus.yding.andersen@regionh.dk (C.Y.A.); 2Department of Clinical Medicine, University of Copenhagen, 2200 Copenhagen, Denmark; simonehildorf@gmail.com (S.H.); dina.cortes@regionh.dk (D.C.); joergen.mogens.thorup@regionh.dk (J.T.); 3Department of Pediatric Surgery, University Hospital of Copenhagen, Rigshospitalet, 2100 Copenhagen, Denmark; 4Centre of Andrology & Fertility Clinic, Department D, Odense University Hospital, 5000 Odense, Denmark; jens.fedder@rsyd.dk; 5Research Unit of Human Reproduction, Institute of Clinical Research, University of Southern Denmark, 5230 Odense, Denmark; 6Department of Pathology, University Hospital of Copenhagen, Rigshospitalet, 2100 Copenhagen, Denmark; erik.clasen-linde@regionh.dk; 7Department of Pediatrics and Adolescent Medicine, Copenhagen University Hospital Hvidovre, 2650 Copenhagen, Denmark

**Keywords:** human immature testicular tissue, cryptorchidism, infertility, organotypic culture, fertility preservation, testicular tissue cryopreservation

## Abstract

Organotypic culture of human fetal testis has achieved fertilization-competent spermatids followed by blastocysts development. This study focuses on whether the organotypic culture of testicular tissue from infant boys with cryptorchidism could support the development of spermatogonia and somatic cells. Frozen-thawed tissues were cultured in two different media, with or without retinoic acid (RA), for 60 days and evaluated by tissue morphology and immunostaining using germ and somatic cell markers. During the 60-day culture, spermatocytes stained by boule-like RNA-binding protein (BOLL) were induced in biopsies cultured with RA. Increased AR expression (*p* < 0.001) and decreased AMH expression (*p* < 0.001) in Sertoli cells indicated advancement of Sertoli cell maturity. An increased number of SOX9-positive Sertoli cells (*p* < 0.05) was observed, while the percentage of tubules with spermatogonia was reduced (*p* < 0.001). More tubules with alpha-smooth muscle actin (ACTA, peritubular myoid cells (PTMCs) marker) were observed in an RA-absent medium (*p* = 0.02). CYP17A1/STAR-positive Leydig cells demonstrated sustained steroidogenic function. Our culture conditions support the initiation of spermatocytes and enhanced maturation of Sertoli cells and PTMCs within infant testicular tissues. This study may be a basis for future studies focusing on maintaining and increasing the number of spermatogonia and identifying different factors and hormones, further advancing in vitro spermatogenesis.

## 1. Introduction

Currently, testicular tissue cryopreservation (TTC) before gonadotoxic treatment has been applied in many countries to preserve fertility in boys diagnosed with childhood cancer or non-malignant diseases [1,2,3,4]. In addition, one in five boys with cryptorchidism, who have undergone orchidopexy in the first year of life, are at risk of compromised fertility in adulthood [5]. In fact, one in five men with azoospermia have a history of cryptorchidism [6], suggesting that an early biopsy containing spermatogonial stem cells (SSCs) obtained for TTC in connection with childhood orchiopexy may benefit the future prospects of fertility [7,8]. TTC from prepubertal boys has been performed for two decades [2], and some of these boys are now reaching the reproductive age underlining the need to develop clinically applicable techniques to use these biopsies and restore their fertility.

Several techniques have been explored to produce sperm from immature testicular tissue (ITT): SSC transplantation, testicular tissue grafting, and in vitro maturation (IVM) of SSCs [9,10]. Completion of in vitro spermatogenesis has been achieved in animals, but none of these strategies succeeded in generating sperm from human ITT. To avoid the potential risk of re-introducing malignant cells to boys who have had cancer, IVM of SSCs is a promising strategy. IVM can be performed by culturing ITT fragments in two-dimensional or three-dimensional culture systems [11,12,13]. So far, the most successful IVM from ITT fragments is an organotypic culture system, which preserves the testicular architecture and paracrine communication [9]. Sato and colleagues firstly reported the successful completion of spermatogenesis from fresh mouse ITT by applying an organotypic culture system [14], which has also successfully been applied to mouse frozen-thawed ITT resulting in the production of healthy offspring [15]. At present, four human studies have reported an organotypic culture of human ITT achieving mature germ cells [13,16,17,18]. However, full spermatogenesis using this technique has not yet been achieved in human ITT. Recently, one promising study demonstrated that the organotypic culture of human fetal testis could generate functional spermatids that can support fertilization and the development of blastocysts [19]. This demonstrates the capacity and plasticity of the fetal SSCs, but which have little clinical relevance because the testicular tissue is obtained from an abortion. Fetal testicular tissue differs from prepubertal testicular tissue in the phenotype of the germ cells present and the level of hormone secretion. Thus, it may be envisioned that improved culture media could stimulate successful in vitro spermatogenesis. The culture media used for the human fetal testis tissue [19] contained a variety of growth factors, including bone morphogenetic protein 4/7 (BMP4/7), stem cell factor (SCF), basic fibroblast growth factor (bFGF) that advanced the differentiation of germ cells [20,21,22,23,24,25], and epidermal growth factor (EGF) and bFGF that improved SSC self-renewal and maintenance [26,27]. Moreover, Activin A stimulates germ cell differentiation and Sertoli cell proliferation [28,29]. Bovine pituitary extract (BPE) was also included containing a variety of hormones, cytokines, mitogens, and growth factors, that supports cell proliferation and protects against oxidative stress [30]. 

In this study, we applied a xeno-free culture media composition similar to that used by Yuan and co-workers [19] with some modifications. In their study, the concentrations of follicle-stimulating hormone (FSH) and testosterone were exceptionally high. Therefore, we reduced the hormone concentrations of FSH and testosterone. Additionally, BPE is not a xeno-free component and cannot be used in media clinically applied. Hence, we excluded BPE in our xeno-free culture media. All the testis biopsies used in our study were obtained from infant boys with bilateral cryptorchidism. We aimed to study whether the organotypic culture of testicular tissue from infant boys with cryptorchidism could support the development and advancement of SSCs and somatic cells.

## 2. Results

### 2.1. Assessment of Serum Hormones

At the time of surgery, all patients had levels of serum FSH, luteinizing hormone (LH), and inhibin B within the normal ranges except for patient #3, who showed a reduced number of germ cells per tubular cross-section (G/T) and low inhibin B reflecting a diminished testicular endocrine function, combined with a low LH suggesting a gonadotropin insufficiency (Table 1). Moreover, testosterone production was below the detection limit (data not shown) in all four patients. However, no correlations were found between hormonal values evaluated at surgery and the results of in vitro maturation.

### 2.2. Histological Analysis of ITT before and after Organotypic Culture

On day 0 (D0, before culture), the most advanced germ cells were spermatogonia present within all ITT biopsies (Johnsen score 3, Table 1, Figure 1A). The number of G/T was reduced already at the time of surgery in two patients (#3 and #4), and the number of spermatogonia A dark per tubular cross-section (AdS/T) was reduced in one patient (#4) (Table 1).

On day 60 (D60), the mean percentage of well-preserved tubules (score 3 and 4) was 88% for medium 1 (M1) and 91% for medium 2 (M2). Both showed a significant decline compared to D0 (both *p* < 0.001), but no significant difference was found between the two culture media (*p* = 0.9) (Figure 1B). After 60-day culture, the lumen of tubules increased. The mean width of tubular cross-sections was 86 μm for M1 and 87 μm for M2, with a significant increase at D60 compared to D0 (both *p* < 0.001) (Figure 1C), indicating tubules were enlarged over the culture period. No statistical difference was observed between the two culture media (Figure 1C).

### 2.3. Spermatogonial Survival and Maturation

There were melanoma antigen genes-A (MAGE-A)/G antigen (GAGE)/VASA-positive spermatogonia before and after the 60-day culture (Figure 2A). From the Periodic acid-Schiff (PAS) staining (Figure 1A), the percentage of tubules containing at least one spermatogonium showed a significant reduction after 60-day culture (Figure 2B,C). The percentage of tubules containing at least one MAGE-A/GAGE/VASA-positive cell significantly decreased in both culture media at D60 compared to D0 (M1: all *p* < 0.001; M2: all *p* < 0.001), and the percentage was higher in culture M1 than M2 (all *p* < 0.001) (Figure 2D,F,H). The number of MAGE-A-positive cells per positive tubule was significantly reduced in both culture media at D60 compared to D0 (*p* = 0.01, *p* = 0.04, respectively) (Figure 2E). The number of GAGE-positive cells per positive tubule was significantly decreased in culture M2 at D60 compared to D0 (*p* = 0.01) (Figure 2G). However, no statistical difference was found between the two culture media (Figure 2E,G,I).

We found boule-like RNA-binding protein (BOLL)-positive cells at D60 in M2 (patient #3) (Figure 2J). The positive control of BOLL (spermatocytes and spermatids) was shown in Appendix A. However, more advanced stages of germ cell development beyond BOLL were not found after the culture (Appendix A). Positive control of related markers was presented in Appendix A.

### 2.4. Sertoli Cell Survival and Maturation

All tubules contained SOX9-positive cells at D0 and D60 (Figure 3A). The number of SOX9-positive cells per tubule was significantly increased in both media at D60 compared to D0 (M1: *p* = 0.001; M2: *p* = 0.004) (Figure 3B), but no significant difference between the two culture media.

A significant reduction in the percentage of anti-Müllerian hormone (AMH)-positive tubules was detected after 60 days of culture in M1 and M2 compared to D0 (both *p* < 0.001) (Figure 3C). Based on different AMH intensities, the percentage of “strong” AMH tubules showed a significant decrease in both media at D60 compared to D0 (M1: *p* < 0.001, M2: *p* < 0.001), a similar result shown in “weak” AMH tubules (M1: *p* < 0.001, M2: *p* < 0.001) (Figure 3D). No significant difference in AMH expression was observed between the two culture media (Figure 3D).

Androgen receptor (AR)-positive Sertoli cells were induced during the 60-day culture (Figure 3E). At D0, there were no AR-positive Sertoli cells within seminiferous tubules. In both media at D60 compared to D0, a significant increase was shown in the percentage of tubules containing AR-positive Sertoli cells (both *p* < 0.001) and the number of AR-positive Sertoli cells (both *p* < 0.001) (Figure 3F,G).

We found that a few tubules showed a disorganized staining pattern of zonula occludens 1 (ZO-1) at D0 in testis biopsies from two patients (patients #1 and #4). After 60-day culture, the disordered distribution of tight junctions stained by ZO-1 was observed in testis biopsies from all patients (Figure 3H). 

Taken together, the expression of AMH decreased and AR increased, and ZO-1 was present during the 60-day culture. 

### 2.5. Proliferation of Spermatogonia and Sertoli Cells

Proliferating spermatogonia and Sertoli cells were detected after 60 days (Figure 4A,B). The percentage of tubules with proliferating spermatogonia and the number of proliferating spermatogonia per tubule increased at D60 in both media compared to D0, but no significant difference was observed (Figure 4C,D). The percentage of tubules with Ki67-positive Sertoli cells showed a substantial increase at D60 irrespective of the media used compared to D0 (M1: *p* < 0.001, M2: *p* < 0.001) (Figure 4E). The number of Ki67-positive Sertoli cells per positive tubule increased significantly at D60 irrespective of the culture media compared to D0 (M1: *p* = 0.01, M2: *p* = 0.03) (Figure 4F). No statistical difference was observed between the two culture media.

### 2.6. Peritubular Myoid Cells

Alpha-smooth muscle actin (α-SMA, also named ACTA)-positive peritubular myoid cells (PTMCs) developed during the 60-day culture (Figure 5A–C). At D0, ACTA-positive PTMCs were only observed in a testicular biopsy from one patient (Patient #1), but at D60, they were shown in testicular biopsies from three patients (Patient #1, #2, #3). A significantly higher percentage of tubules with ACTA-positive PTMCs was observed at D60 in M1 compared to D0 (*p* = 0.02) (Figure 5D). More tubules expressed ACTA-positive PTMCs after culture in M1 than in M2 (*p* = 0.02) (Figure 5D).

### 2.7. Leydig Cell Functionality

At D0, there were no steroidogenic acute regulator (STAR)-positive Leydig cells (Figure 6A), whereas, at D60, STAR-positive Leydig cells were observed (Figure 6B). The cytochrome P450 17A1 (CYP17A1) positive Leydig cells were detected both at D0 and D60 irrespective of the culture media (Figure 6C,D), indicating that Leydig cells continued steroidogenic function during the 60-day culture.

## 3. Discussion

To our knowledge, this is the first study to perform an organotypic culture of testicular tissues from infant boys with bilateral cryptorchidism for 60 days to achieve in vitro maturation of germ and somatic cells. We found that the testicular structure was maintained, and tubules were enlarged during the 60-day culture. The most advanced stage of germ cells achieved during the 60-day culture were spermatocytes. Meanwhile, maturation of somatic cells, including Sertoli cells and PTMCs, was initiated, and steroidogenic activity was sustained.

Overall, the number of spermatogonia was reduced, similar to previous studies [13,17,18]. Portela and colleagues hypothesized that testicular tissues with a higher Johnson score before culture would have a higher sensitivity to in vitro conditions [18]. In our study, testicular samples were from infant boys younger than 1.5 years old with a Johnson score of 3, and the most advanced germ cells were spermatogonia before culture. Our results demonstrated the survival of spermatogonia with proliferative capacity, but the number of spermatogonia was considerably reduced during the culture period. We found that the number of different SSC phenotypes (MAGE-A, GAGE, VASA) were all reduced. Compared to a previous study [13] which included two 2-year-old boys and three older prepubertal boys diagnosed with malignant tumors, the percentage of well-preserved tubules (integrity scores 3 and 4), around 80–90%, was similar to ours. The number of MAGE-A-positive cells per tubule and the number of Ki67-positive spermatogonia per tubule were similar to previous studies [13,16]. Therefore, the addition of bFGF and EGF to the media used in our study was unable to significantly promote spermatogonia self-renewal. 

We found BOLL-positive spermatocytes in biopsies cultured in M2 (with RA), contrary to previous studies [13,19], where meiosis advanced when biopsies were cultured in a medium without RA, which is considered a factor advancing germ cell meiosis [31,32]. It’s reported that RA could result in severe cell apoptosis and destruction of tubule morphology [19], but we did not find a difference in tubule morphology in relation to the presence of RA. This may relate to the small number of biopsies included, but we found that the percentage of tubules with spermatogonia and ACTA-positive expression was significantly higher in M1 (without RA) than in M2. The exact role of RA in advancing spermatogenesis in vitro requires further investigations. The BOLL-positive spermatocytes, found only in testicular tissues from one patient (patient #3), could be because we only stained four sections at different depths/patient/medium for the BOLL marker. Although Patient #3 (the only individual) had reduced G/T, low inhibin B, and a gonadotropin insufficiency at the time of surgery, we were unable to determine the effect of G/T and endogenous hormones on in vitro maturation as only four patients were included in this study. At present, de Michele and colleagues have developed haploid germ cells from prepubertal boys [13]; Yuan and colleagues have achieved fertilization-competent spermatids from fetal testis [19], and we achieved BOLL-positive spermatocytes by using a modified media used in Yuan and colleagues’ study (18). The concentrations of FSH and testosterone (according to the FSH from human pituitary (Sigma-Aldrich, St. Louis, MO, USA), the specific activity of FSH is 7000 IU/mg, therefore, FSH: 200 ng/mL = 1400 IU/L; testosterone: 10 mM)) were much higher in Yuan and colleagues’ study [19] as compared to de Michele and colleagues’ study (FSH: 5 IU/L; without testosterone) and ours (FSH: 10 IU/L; testosterone: 10 μM). Previous reports showed that a higher concentration of FSH (50 IU/L) did not induce meiosis, whereas FSH at 5 IU/L did, based on testicular samples from prepubertal boys [13,16]. It’s reported that a higher percentage of haploid cells developed with FSH concentrations increasing from 10 IU/L to 50 IU/L but no difference from 50 IU/L to 100 IU/L based on adult testicular samples [33]. The optimal concentrations of FSH and testosterone applied to human prepubertal testicular tissue to induce spermatogenesis in vitro need closer studies.

In addition, we found some seminiferous tubules that developed the blood-testis barrier (BTB), however, in a disorganized pattern during the culture period, similar to that previously reported [34]. The disorganized BTB, which may be unable to separate the seminiferous epithelium into basal and adluminal compartments, might be inadequate to support full spermatogenesis in vitro.

Taken together, there is no consensus on the optimal concentration of FSH and testosterone in the culture medium to advance meiosis in vitro. Since testosterone synergizes with FSH on spermatogenesis, the different testosterone concentrations could exert different synergistic effects. Although the intratesticular high concentration of testosterone is not required for full spermatogenesis in adult men [35], the effect of high testosterone concentration on in vitro spermatogenesis from immature human testicular tissue might differ. Further studies on the effect of different concentrations of FSH and testosterone on human immature testicular biopsies are needed.

BPE, which contained a mixture of different hormones, cytokines, and growth factors, was not required for complete in vitro spermatogenesis in mouse ITT [14,36]. We did not include BPE in our media in contrast to the study done by Yuan and colleagues [19] and whether BPE is required for full in vitro spermatogenesis in the culture of human ITT requires further studies. 

After 60-day culture, all tubules contained SOX9-positive Sertoli cells. The mean number of SOX9-positive Sertoli cells per tubular cross-section was 37 in our study and higher than previously reported in the study by Medrano and colleagues, in which the number was around 20 per tubular cross-section after 14-day culture and around 10 per tubular cross-section after 70-day culture [17]. This is most likely related to the proliferative ability of Sertoli cells in our study. It has been reported that BMP4 and Activin A promoted the proliferation of Sertoli cells [37,38,39], and these growth factors were included in our media. We found that Sertoli cells became more mature during the culture period, and the increased number of Sertoli cells suggested that the addition of BMP4 and Activin A should be included in future studies. 

Previous studies did not show a significantly increased AR expression during the culture period [13,17,18]. This could be related to the age of the patients. Portela and colleagues reported that the presence of AR expression by Sertoli cells was already lost during the first week of culture when testicular tissue from prepubertal boys aged 10 and 14 was used [18]. In our study, including testicular tissue taken from infant boys, there were no AR-positive Sertoli cells at the start of the culture period. However, expression of AR was achieved during the culture period reflecting that the composition of the medium or that the age of boys may affect the ability of Sertoli cells to mature in vitro. 

In our study, the composition of the culture media appeared to induce maturation of Sertoli cells, but more and better maturation is required to support full spermatogenesis. The dynamic addition of growth factors to keep the balance between proliferation and maturation of Sertoli cells might be a goal for future studies.

Increased expression of ACTA indicated that our culture media induced the development of smooth muscle characteristics of PTMCs. Medrano and colleagues showed similarly that ACTA expression was significantly increased at RNA level on days 14 and 70 by culturing human immature testicular tissues [17]. ACTA expression, which commences around puberty, is mainly induced by androgens [40,41]. Therefore, the increased expression of ACTA indirectly indicated that Leydig cells maintained their function to secrete androgens during the culture period. Also, the diameter of the tubules in our study increased from 60 μm at D0 to an average of 86 μm at D60, paralleling the development of the lumen. However, the diameter was still lower than the minimum diameter of tubules supporting full spermatogenesis in adult testis (i.e., 110 μm) [42]. The positive expression of CYP17A1 and STAR in Leydig cells further demonstrated a continued steroidogenic capacity of Leydig cells during the culture period. Considering the decrease of fetal Leydig cells and the slow increase of adult Leydig cells in the testis from these infant boys, it might support that more testosterone needs to be added to the culture media for inducing in vitro spermatogenesis from infant testicular tissue.

In our culture media, we used human umbilical cord plasma, which contained a mixture of growth factors, such as insulin-like growth factor-1, EGF, bFGF, and cytokines [43,44]. However, 2% of human umbilical cord plasma could not support full spermatogenesis from infant testicular tissue. Umbilical cord plasma qualifies as a protein additive that is xeno-free, and further evaluation, including testing of different concentrations, should be attempted. 

The scarcity of available testicular tissue from infant boys and the small size of testicular biopsies were limitations of this study. Hence, we did not include more time points in this study. Further studies could apply various time points along with other makers to provide more detailed information on the development of germ cells and somatic cells during in vitro conditions.

In conclusion, in boys with cryptorchidism, our organotypic culture conditions maintained testicular structure during a 60-day culture period. The number of spermatogonia was reduced, but spermatogonia matured to the spermatocytes stage. Further, Sertoli cells and PTMCs also advanced maturation during the culture period. Therefore, the present study demonstrates the developmental potential of the SSCs in infant boys who suffer from cryptorchidism and warrants studies, potentially including a prolonged culture period and using various factors and hormones to further advance meiosis.

## 4. Materials and Methods

### 4.1. Human Testicular Tissue

Testicular biopsies were obtained from four infant boys (0.5–1.4 years) with bilateral cryptorchidism who underwent orchidopexy at the Department of Pediatric Surgery, Rigshospitalet, Denmark (Table 1). One of the patients (#2) had a reduced birth weight of 2.2 kg. None of the patients received hormonal treatment. Testicular tissue was obtained from an incision in the tunica albuginea and divided into three fragments (Figure 7). One fragment was immersed in Stieve’s fixative and sent to the Department of Pathology for pathological assessment, as previously described [45]. The remaining two fragments were placed in McCoy 5A medium (modified 22330-021, Gibco, Paisley, UK) for transportation to the laboratory (10 min. transport), where TTC was performed for future clinical (fertility preservation biobank) and research purposes. The fragments for TTC were equilibrated in 1.5 M ethylene glycol, 0.1 M sucrose, and 10 mg/mL human serum albumin (HSA) for 20 min, followed by a slow-freezing procedure and storage in liquid nitrogen as previously described [46].

### 4.2. Hormonal Evaluation

The concentrations of serum hormones, including FSH, LH, and inhibin B, were assessed. Serum FSH and LH were evaluated by a time-resolved immunofluorometric assay (Delfia, Wallac, Turku, Finland) and inhibin B by a specific two-sided enzyme immunometric assay (Inhibin B gen II, Beckman Coulter Ltd., High Wycombe, UK). Inhibin B values below the normal age-related 2.5 percentile were considered reduced.

### 4.3. Culture Media and Method

The study included two types of xeno-free culture media. M1: Minimum Essential Medium-alpha (MEM-α) (12571-063, Gibco, Bleiswijk, Netherlands), 2% human umbilical cord plasma (70020, Stemcell Technologies, Kent, WA, USA), 10% KnockOut SR XenoFree CTS (KSR) (12618-012, Gibco, Bleiswijk, The Netherlands), recombinant human glial cell line-derived neurotrophic factor (GDNF) (20 ng/mL, G1777, Sigma, Steinheim, Germany), bFGF (20 ng/mL, 234-FSE, R&D, Oxford, UK), EGF (20 ng/mL, E9644, Sigma, Steinheim, Germany), SCF (20 ng/mL, 300-07-100UG, Peprotech, Rocky Hill, NJ, USA), BMP4 (20 ng/mL, 120-05ET 100UG, Peprotech, Rocky Hill, NJ, USA), Activin A (100 ng/mL, 120-14P-50UG, Peprotech, Rocky Hill, NJ, USA), FSH (10 IU/L, Rekovelle, Ferring, Hvidovre, Denmark), testosterone (10 μM, 86500, Sigma, Steinheim, Germany), 0.5% penicillin-streptomycin (15140122, Gibco, Bleiswijk, The Netherlands). M2 consisted of the same components as M1 plus 10 μM retinoic acid (RA) (R2625-100MG, Sigma, Steinheim, Germany).

Before culture, vials with the cryopreserved tissue were thawed in a 37 °C water bath. Immediately after thawing, the testicular tissue was placed in 0.75 M ethylene glycol, 0.25 M sucrose, 10 mg/mL HSA in PBS for 10 min, and moved to 0.25 M sucrose in PBS and 10 mg/mL HSA for 10 min, and finally to PBS and 10 mg/mL HSA for 10 min. After removal of cryoprotectants, each testicular tissue (approximately 3 × 3 × 3 mm^3^) was cut into 1–1.5 mm^3^ fragments and positioned on a 0.35% (*w*/*v*) agarose gel stands (10 × 8 × 6 mm) (16500-100, Invitrogen, Carlsbad, CA, USA). To replace the water inside the gel with culture media, the gel stands were incubated in the culture media for at least 24 h before the tissue fragment was loaded on top and placed in a 4-well dish. The culture media covered the gel stands but not the tissue leaving an air-liquid interface for the tissues, and were changed every two days. Testicular tissues were cultured at 34 °C in a humid atmosphere with 5% CO_2_. Yuan and colleagues [19] induced the development of spermatids in fetal testicular tissues during a 50-day culture period, shorter than the length of spermatogenesis in vivo. De Michele et al. reported that the development of haploid cells was induced in prepubertal testicular tissues during 64- and 139-day culture periods [13]. A 60-day culture period represents a compromise between the difficulty in maintaining long-term cultures and the ability to demonstrate the advancement of meiosis in germ cells.

### 4.4. Staining before and after Culture

The tissue fragment for pathological assessment was included in this study as control tissue without culture as a day 0 (D0) sample. This tissue fragment was evaluated with routine IHC markers for assessing the germ cell number of each patient according to the studies by Hildorf et al. [45,47] and underwent the same staining as the cultured tissues described below.

The cultured tissues were collected at D60 and fixated with Bouin fixative. The fixed tissues were deparaffinized and rehydrated with a series of graded ethanol, embedded in paraffin, and cut into 5-μm serial sections. PAS and IHC staining were performed on two independent sections with an interval of at least 50 μm.

For IHC staining, sections underwent antigen retrieval in a TEG buffer (10 mM Tris, 0.5 mM ethylene glycol-bis 2-aminoethylehter)-*N*,*N*,*N*′,*N*′-tetraacetic acid (EGTA), pH 9) for 30 min. Endogenous peroxidases were blocked by 0.5% H_2_O_2_ for 15 min, and non-specific binding sites were blocked by 4% bovine serum albumin (BSA) and 5% donkey serum (DS)/rabbit serum (RS) for 30 min at room temperature (RT). Sections were incubated at 4 °C overnight with primary antibodies. The primary antibodies (Appendix A) were diluted in 4% BSA and 5% DS/RS buffer: mouse anti-MAGE-A monoclonal antibody (1:200), mouse anti-GAGE monoclonal antibody (1:150), goat anti-VASA polyclonal antibody (1:100), rabbit anti-SOX9 polyclonal antibody (1:100), goat anti-AMH polyclonal antibody (1:100), rabbit anti-AR monoclonal antibody (1:100), rabbit anti-ZO-1 polyclonal antibody (1:200), rabbit anti-ACTA polyclonal antibody (1:150), mouse anti-Ki67 monoclonal antibody (1:50), rabbit anti-BOLL polyclonal antibody (1:100), rabbit anti-ACROSIN monoclonal antibody (1:100), goat anti-CREM polyclonal antibody (1:500), rabbit anti-PRM1 monoclonal antibody (1:200), goat anti-CYP17A1 polyclonal antibody (1:200), mouse anti-STAR polyclonal antibody (1:200).

After washes in Tris-buffered saline (TBS)/Tween, the sections were added with the secondary antibody donkey/rabbit anti-mouse/rabbit/goat horseradish peroxidase (HRP) (Dako, Glostrup, Denmark) for incubating 30 min at RT, visualized with 3,3′-diaminobenzidine tetrahydrochloride (Dako) for 1–2 min, counterstained with Mayer’s hematoxylin, and mounted with Pertex (00801, Histolab, Gothenburg, Sweden). The negative control was performed using universal negative control serum (NC498H, Biocare Medical, Hague, The Netherlands). Mature human testicular tissue was used for positive control. Sections were evaluated and imaged through a Zeiss microscope with a Leica digital microscope camera.

### 4.5. Tissue Histological and Immunohistochemical Assessment

#### 4.5.1. Germ Cell Density before Culture

We evaluated the germ cell density based on the D0-sections according to the assessment previously described [45,47]. For each patient, the mean number of G/T and the mean number of AdS/T were evaluated on at least 100 and 250 cross-sectioned tubules, respectively, as previously described [47]. G/T was considered normal when the value was above the lower range interval established by previously published normal materials, and the lower range of normal AdS/T was to be 0.01 [47]. Moreover, sections were evaluated with the Johnsen score [48].

#### 4.5.2. Integrity and Diameter of Seminiferous Tubules

We evaluated the integrity of all seminiferous tubules according to a previously described scoring system [13]. In brief, there were four scores (1 to 4, with 4 being the best) related to four parameters that consisted of cell adhesion to the basement membrane, cell cohesion, less than 5% pyknotic nuclei, and clear distinction of germ cells and Sertoli cells. The diameter of tubules was measured on 10 tubular cross-sections per patient/medium.

For the following assessment of spermatogonial survival and intratubular cell proliferation, germ cell maturation, and somatic cell maturation, only well-preserved tubules with an integrity score of 3 or 4 were evaluated. The total number of tubules per section was evaluated. Each section contained at least 40 seminiferous tubules.

#### 4.5.3. Spermatogonial Survival and Intratubular Cell Proliferation

We used antibodies against MAGE-A, GAGE, and VASA to identify spermatogonia. The number of tubules with MAGE-A/GAGE/VASA-positive germ cells and a total number of positive germ cells per section were counted to quantify spermatogonial survival. We used the Ki67 marker to detect the proliferating spermatogonia and Sertoli cells. The number of Ki67-positive germ cells and Sertoli cells per tubule was counted according to the distinct morphology of germ cells and Sertoli cells.

#### 4.5.4. Germ Cell Maturation

The spermatocyte marker BOLL demonstrated the meiotic state of germ cells. The spermatid markers, ACROSIN, CREM, and PRM1, represented the post-meiotic state of germ cells.

#### 4.5.5. Somatic Cell Maturation

SOX9-positive Sertoli cells were counted in 10 tubular cross-sections/patient/medium. Immature Sertoli cells were detected with the marker AMH, and the intensity of expression was monitored. Because all tubular cross-sections contained SOX9-positive Sertoli cells while part of tubular cross-sections contained AR-positive Sertoli cells, there were not enough 10 tubular cross-sections for evaluation with AR-positive expression. Therefore, the number of tubules with AR-positive Sertoli cells and the number of AR-positive cells within 5 positive tubular cross-sections/patient/medium were counted. ZO-1, a marker for tight junctions between Sertoli cells, was used to show the formation of the BTB. To evaluate PTMCs, the number of tubules per section with ACTA-positive PTMCs was counted. To assess the steroidogenic activity of Leydig cells, we performed IHC staining for CYP17A1 and STAR.

### 4.6. Statistical Analysis

Kruskal-Wallis test with Dunn’s multiple comparisons test was used to analyze the number of spermatogonia/MAGE-A/GAGE/VASA/SOX9/AR/Ki67-positive cells and tubular diameter. Chi-square test was used to analyze tubular integrity and the percentage of tubules with the expression of spermatogonia/MAGE-A/GAGE/VASA/AMH/AR/Ki67/ACTA at D60 compared to that at D0. *p* values < 0.05 were considered statistically significant. Results were shown as mean ± sd. GraphPad Prism version 8.0 and SPSS version 23.0 were used for statistical analyses.

## Figures and Tables

**Figure 1 ijms-23-07975-f001:**
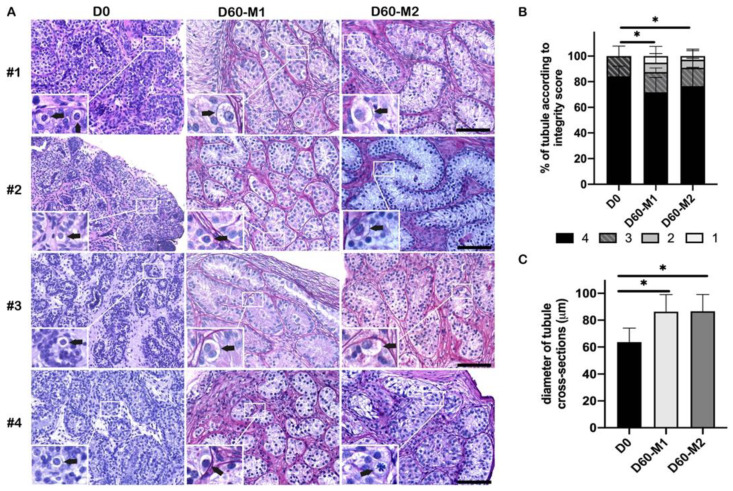
Assessment of ITT organotypic culture. (**A**) The histology of ITT before (D0) and after (D60) organotypic culture (M1 (without RA), M2 (with RA)). Black arrows in the inserts indicated spermatogonia. Scale bar = 100 μm. (**B**) Percentage of seminiferous tubules according to integrity score. A significant decrease was observed in the percentage of well-preserved tubules (integrity scores 3 and 4) at D60 in both culture media compared to D0 (both *p* < 0.001). No difference was shown between the two culture media. (**C**) The diameter of tubular cross-sections. Asterisks indicated statistically significant differences.

**Figure 2 ijms-23-07975-f002:**
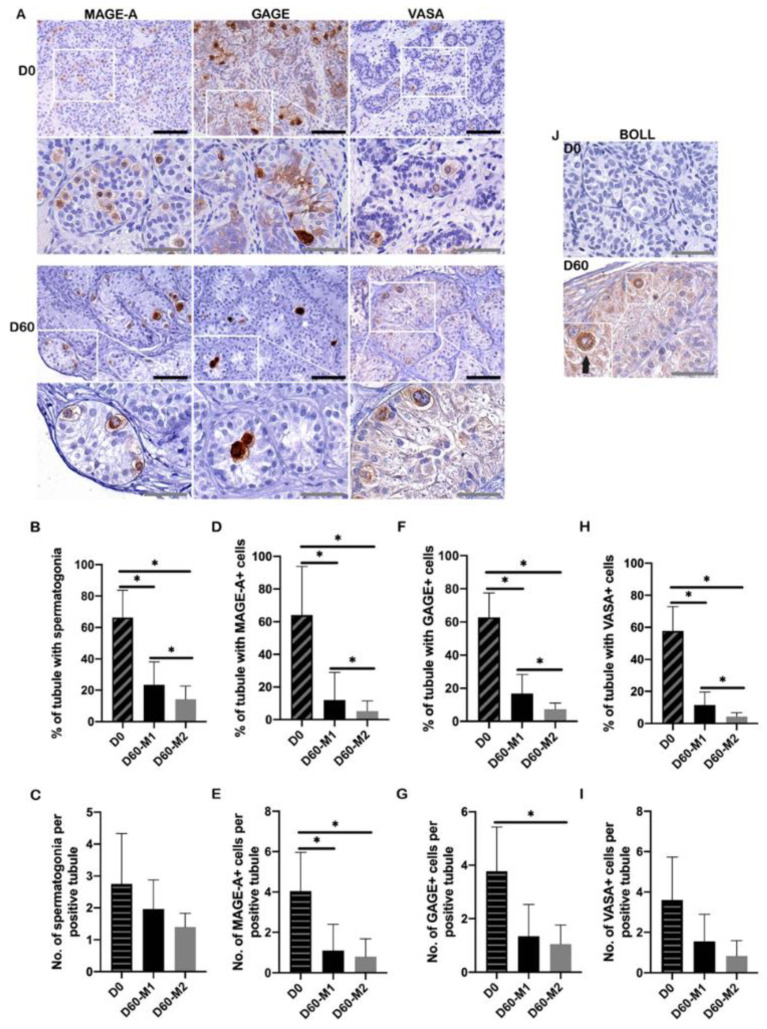
Spermatogonia survival and maturation. (**A**) Immunohistochemical staining of three spermatogonial markers (MAGE-A, GAGE, VASA) within ITT at D0 and D60. The brown color indicated MAGE-A/GAGE/VASA-positive germ cells. Scale bars: black = 100 μm; grey bar = 50 μm. (**B**) Percentage of tubules with spermatogonia (positive tubule) according to its morphology (PAS staining). (**C**) The number of spermatogonia per positive tubule. (**D**) Percentage of tubules with MAGE-A-positive spermatogonia (positive tubule). (**E**) The number of MAGE-A-positive spermatogonia per positive tubule. (**F**) Percentage of tubules with GAGE-positive spermatogonia (positive tubule). (**G**) The number of GAGE-positive spermatogonia per positive tubule. (**H**) Percentage of tubules with VASA-positive spermatogonia (positive tubule). (**I**) The number of VASA-positive spermatogonia per tubule (positive tubule). (**J**) Immunohistochemical (IHC) staining of BOLL. Positive expression of BOLL after 60-day culture was only observed in M2. The black arrow indicated BOLL positive cell. Scale bar = 100 μm. Asterisks indicated statistically significant differences.

**Figure 3 ijms-23-07975-f003:**
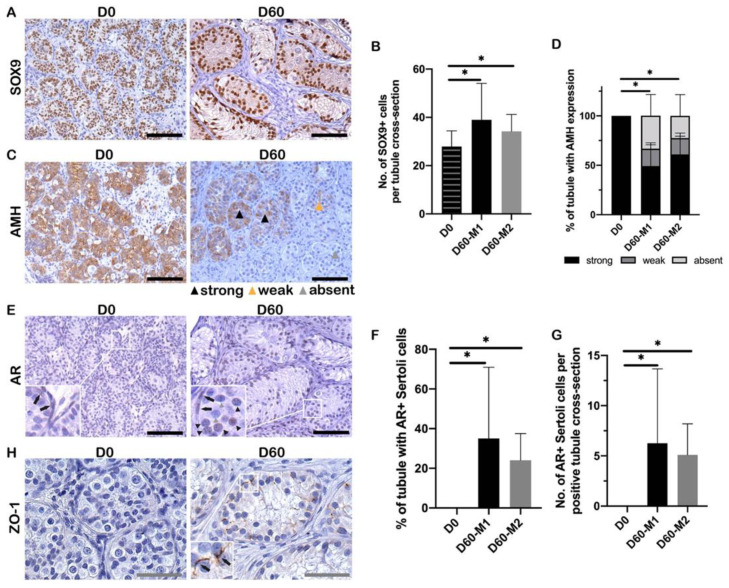
Sertoli cell survival, maturation, and functionality. (**A**) SOX9 positive staining within ITT at D0 and D60. (**B**) The number of SOX9+ Sertoli cells per tubule before and after 60-day organotypic culture. (**C**) AMH positive staining within ITT at D0 and D60. Black triangle: strong staining of AMH; orange triangle: weak staining of AMH; grey triangle: absent staining of AMH. Scale bar = 100 μm. (**D**) Percentage of tubules with immunohistochemical staining of AMH. The staining of AMH was evaluated by a semiquantitative score according to the staining color intensity—absent, weak, and strong. The percentage of “strong” AMH tubules showed a significant decrease in both media at D60 compared to D0 (M1: *p* < 0.001, M2: *p* < 0.001), the similar result for “weak” AMH tubules (M1: *p* < 0.001, M2: *p* < 0.001). No significant difference was observed between the two culture media. Scale bar = 100 μm. (**E**) AR staining within ITT at D0 and D60. Black arrowhead: AR-positive Sertoli cells; grey arrowhead: AR-negative Sertoli cells; black arrow: AR-positive peritubular myoid cells (PTMCs). Scale bar = 100 μm. (**F**) Percentage of tubules with AR-positive Sertoli cells (positive tubule) within ITT. (**G**) The number of AR-positive Sertoli cells per positive tubule. (**H**) ZO-1 staining within ITT at D0 and D60. The black arrow indicated ZO-1 positive expression. Scale bar = 50 μm. Asterisks indicated statistically significant differences.

**Figure 4 ijms-23-07975-f004:**
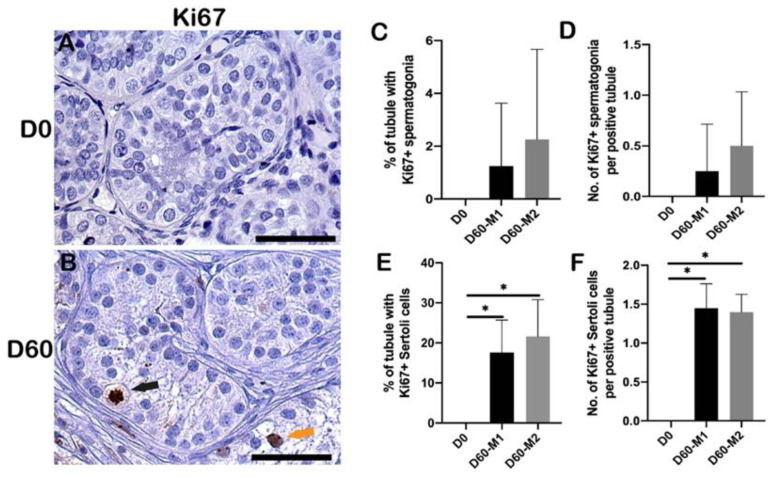
Proliferation of spermatogonia and Sertoli cells. Ki67 positive staining in spermatogonia and Sertoli cells within ITT at D0 (**A**) and D60 (**B**). Black arrow: Ki67-positive spermatogonium; orange arrow: Ki67-positive Sertoli cells. Scale bar = 50 μm. (**C**) Percentage of tubules with proliferating spermatogonia (positive tubule). (**D**) The number of proliferating spermatogonia per positive tubule. (**E**) Percentage of tubules with proliferating Sertoli cells (positive tubule). (**F**) The number of proliferating Sertoli cells per positive tubule. Asterisks indicated statistically significant differences.

**Figure 5 ijms-23-07975-f005:**
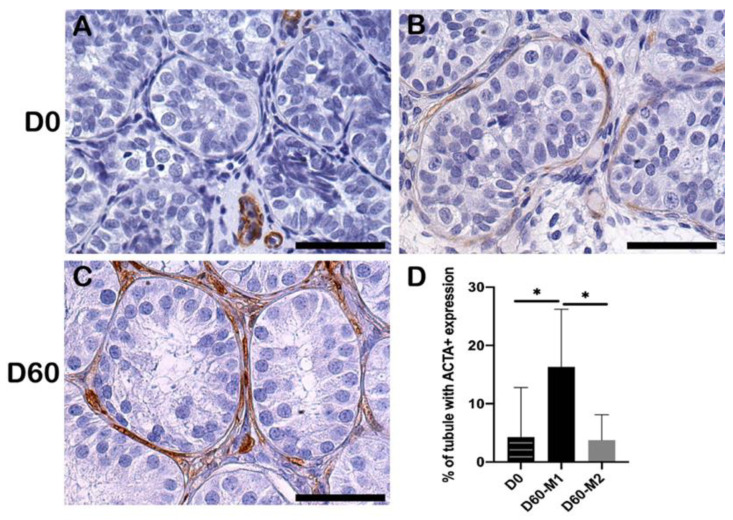
Alpha-smooth muscle actin (ACTA) expression at PTMCs. (**A**) ACTA showed negative expression in PTMCs at D0 and ACTA+ expression on blood vessels. Positive expression of ACTA in PTMCs at D0 (**B**) and D60 (**C**). Scale bar = 50 μm. (**D**) Percentage of tubules with ACTA positive PTMCs at D0 and D60. Asterisks indicated statistically significant differences.

**Figure 6 ijms-23-07975-f006:**
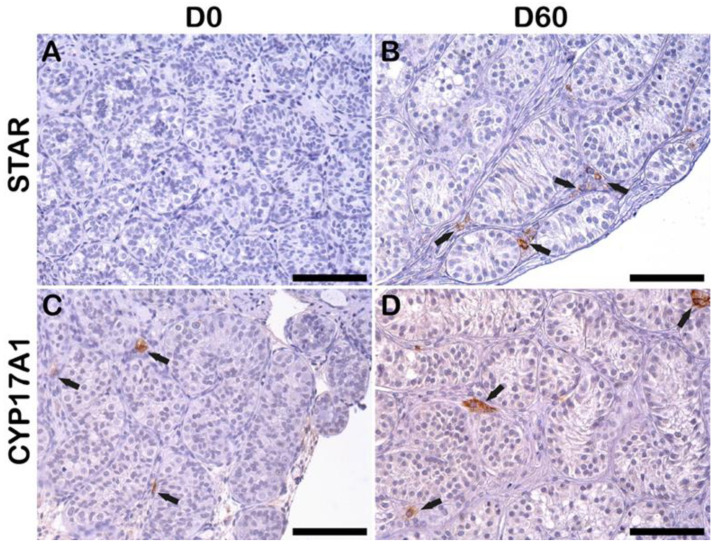
Leydig cell functionality. IHC staining of Leydig cells for STAR at D0 (**A**) and D60 (**B**), CYP17A1 at D0 (**C**), and D60 (**D**). Scale bar = 100 μm. The black arrow indicated positive expressions.

**Figure 7 ijms-23-07975-f007:**
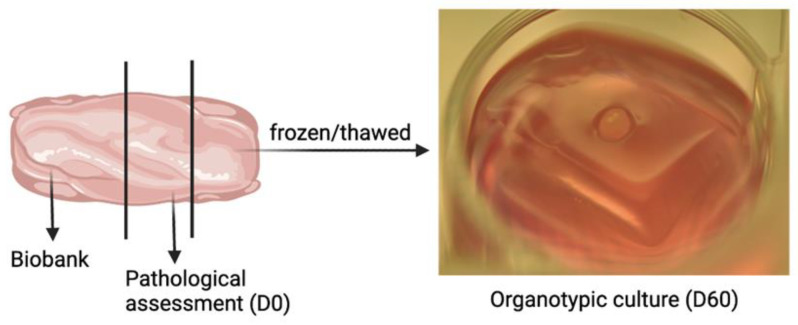
Organotypic culture of immature testis tissue (ITT) from cryptorchid boy. The testis tissue was cut into three fragments for biobank, pathological assessment, and organotypic culture use. The tissue fragment (1–1.5 mm^3^) was placed on an agarose gel stand with an air-liquid interface. D = day.

**Table 1 ijms-23-07975-t001:** Clinical and experimental parameters of infant boys with bilateral cryptorchidism.

Patient ID	Age at Orchidopexy (Year)	Birth Weight (g)	Testis Location *	Serum FSH (IU/L)	Serum LH (IU/L)	Serum Inhibin B (pg/mL)	G/T Mean	AdS/T Mean	Johnsen Score
#1	0.5	3500	abdominal	1.27	1.18	264	2.88	0.08	3
#2	1.4	2200	supra-scrotal	1.49	0.36	105	0.96	0.017	3
#3	1.0	3520	inguinal	0.89	0.05	77	0.28	0.012	3
#4	0.5	4110	annulus externus	0.57	0.85	280	0.69	0.004	3

* Testis location was the location of the undescended testis, which had biopsy for research use.

## Data Availability

The data supporting the findings of this study are available from the corresponding author on a reasonable request.

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
