# Peer review of "Organotypic Culture of Testicular Tissue from Infant Boys with Cryptorchidism"

_ijms, 2022, doi:10.3390/ijms23147975_

Round 1
Reviewer 1 Report
In the manuscript “Organotypic culture of testicular tissue from infant boys with cryptorchidism”, the authors proposed to study whether organotypic culture of testicular tissue from infant boys with cryptorchidism could support the development and advancement of spermatogonial stem cells (SSCs) and somatic cells. The authors conclude that the testicular structure was maintained and tubules were enlarged during the 60-day culture. This is a very interesting paper. Although it seems a bit preliminary, the data is of great interest in the field and contributes to the literature for comparison purposes. Nevertheless, there are many more limitations than the small size of testicular biopsies. For instance, other markers could be done and/or time points. The PTMCs could have been better addressed. Hormonal contribution to the studied process should also be put on spotlight and at least briefly discussed. Overall the paper is good but needs some minor work before it can be considered for publication.
Author Response
Dear Reviewer,
Please see the attachment about the response to your comments. Thank you.
Kind regards,
Danyang Wang

Reviewer 2 Report
In this study, authors aimed to study whether organotypic culture of testicular tissue from infant boys with cryptorchidism could support the development and advancement of SSCs and somatic cells. The study overall is very interesting and well-conducted/written. Introduction is very well-structured and comprehensive; methods are clearly presented, and authors comprehensively discussed their results and limitations of the study. This represents an important study in basic research which may help developing relevant clinical applications in the future. For this reason, I recommend publication after minor revision. Please see below specific comments.
Abstract: I don’t get what BOLL and ACTA mean. Maybe it can be reported they are marker for the germ cell state.
In the introduction, about ref 19, the authors wrote “[19], which demonstrates the capacity and plasticity of the fetal SSCs, but which have little clinical relevance”. Considering that their study is based on a culture media similar to the one used in ref 19, authors should explain why the previous study had “little clinical relevance”.
Methods are at the end of the article. Please, double check all the abbreviation. They should be spell out the first time they appear in the text.
In Methods, section 4.4. refer to the “studies” (plural) of Hildorf et al.
Considering that the length of spermatogenesis is around 3 months, why did author collect the cultured tissues at day 60?
Also, can authors better explain why they decided not to use BPE in the culture media?
Author Response

(The authors gave the same response as above.)
